# Responses Presented by Adult Patients with COVID-19, Based on the Formulated Nursing Diagnoses: A Scoping Review

**DOI:** 10.3390/ijerph19106332

**Published:** 2022-05-23

**Authors:** Vanessa Cortinhal, António Pereira, Sofia Correia, Sérgio Deodato

**Affiliations:** 1Centro Hospitalar Barreiro Montijo EPE, Av. Movimento das Forças Armadas, 2830-003 Barreiro, Portugal; 2Center for Interdisciplinary Health Research, Universidade Católica Portuguesa, 1649-023 Lisbon, Portugal; antonio.pereira@hgo.min-saude.pt (A.P.); sofiacorreia@ucp.pt (S.C.); sdeodato@ucp.pt (S.D.); 3Hospital Garcia de Orta, Av. Torrado da Silva, 2805-267 Almada, Portugal; 4Centro Hospitalar Lisboa Ocidental EPE, Estrada do Forte do Alto do Duque, 1449-005 Lisboa, Portugal; 5School of Nursing, Institute of Health Sciences, Universidade Católica Portuguesa, 1649-023 Lisbon, Portugal

**Keywords:** nursing diagnoses, coronavirus, human responses, nursing, adult patients

## Abstract

(1) Background: this review aims to identify the human responses exhibited by adult patients with COVID-19, by listing the corresponding nursing diagnoses. Nursing diagnosis it’s a clinical analysis of human responses to a person, family, or community. Therefore, it is possible to state that nursing diagnoses represent human responses. (2) Methods: a scoping review was conducted following recommendations provided by the Joanna Briggs Institute (JBI) and the research was carried out between December 2020 and 15 January, 2021, via CINAHL Complete, Complementary Index, MEDLINE, Science Direct, Academic Search Complete, Science Citation Index, Directory of Open Access Journals, Scopus, Social Sciences Citation Index, Business Source Complete, eBook Index (by B-on), and the Cochrane Database of Systematic Reviews (by Cochrane Library). (3) Results: with respect to studies using the NANDA-I taxonomy, the findings have shown that “impaired gas exchange” was the most highlighted nursing diagnosis. ICNP taxonomy, the relevant nursing diagnosis is “cough present”. (4) Conclusions: concurrently, as suggested by the human responses documented in this review, throughout the pandemic, the requirements for adequate care provision have been constantly updated, to improve the quality of life of those patients, as much as possible.

## 1. Introduction

Within a multidisciplinary team, the nurse assumes the role of observing the patient from a holistic point of view [1], focusing on basic human needs [2]. To that purpose, it is essential to formulate a nursing process capable of providing human responses. Iyer, Taptich, and Bernocchi-Losey (1997) conceived human responses as “the way in which the patient responds to a state of health or illness [...]”, referring to the individual’s feelings, perceptions, behaviors, and physiological reactions [3]. Such responses arise from a human being’s needs, the problems related to changes from health to illness, and the relationship with the surrounding environment [4]. The focus of nursing can, thus, be perceived as the course towards human responses to life processes and health problems [5]. In this sense, it becomes crucial to use a standardized language in nursing, embedded in the nursing process, when creating nursing diagnoses [3].

According to Fry (1953), “the first major task in our creative approach to Nursing is to formulate a nursing diagnosis and design a plan which is individual, and which evolves as a result of a synthesis of needs” [6]. According to ISO 18104:2014, a nursing diagnosis is “a label assigned to an assessment finding, event, situation or other health issue, to indicate that it is considered to be noteworthy by the nurse and, where possible, the subject of care” [7]. Concurrently, ICNP (2009, p. 14) regards nursing diagnosis as “a label given by a nurse who makes a decision about the patient or client following assessment” [8]. In addition, according to the NANDA-I taxonomy (2013), nursing diagnosis is “a clinical judgment concerning a human response to health conditions/life processes, or a vulnerability for that response, by an individual, family, group or community” [9].

It is clear that standardized nursing taxonomies ensure an adequate organization and structuring of the human responses identified by the nurse [10], and also that nursing diagnoses are essential when planning care and documenting human responses [11]. In this sense, a proper identification, planning, and resolution of a given situation can only be considered after a clear and accurate assessment by the nurse. In fact, substantiated nursing diagnoses will contribute to interventions which are appropriate for each patient and focused on each individual need [11,12]. Furthermore, their correct documentation allows reinforcing the nurse’s focus on the human responses, providing visibility, and strengthening the stimulus for the continuous improvement of nursing care [10]. Likewise, the thoughtful identification of nursing diagnoses will contribute to increasing care effectiveness, the quality of nursing interventions and its expected outcomes, and, consequently, to reducing hospital costs [10,11].

After presenting a deeper insight into nursing diagnoses, it is important to clarify the use of standardized taxonomies and classifications, since, despite the taxonomies’ differences, the definitions they comprise express concordant ideas. The ISO 18104:2003 standard—later updated by ISO 18104:2014—regulates the language employed in nursing diagnoses, with the purpose of making them safer and, consequently, more efficient. To achieve this, both the ICNP language and the NANDA-I taxonomy were updated in accordance [7,13].

The present review seeks to reply to the following research question: based on the formulated nursing diagnoses, which human responses are exhibited by adult patients with COVID-19? To answer this question, first, it is relevant to clarify the meaning of the terms COVID-19, SARS-CoV-2, 2019-nCoV, and coronavirus [14,15,16], since these were employed as DeCS and/or MeSH descriptors, when performing the database searches. SARS-CoV-2 (also called 2019-nCoV, initially) is the newly discovered coronavirus that started spreading amongst the human population in December 2019, in Wuhan, China. COVID-19 is the infectious disease caused by the virus, and is commonly associated with symptoms, such as fever and extensive lung lesions. In March 2020, the World Health Organization declared the spreading of this virus as a pandemic [14,15,16]. During this outbreak, nursing care has presented itself as a key element in healthcare systems worldwide, due to the competence of nurses in diagnosing various issues related to patients with COVID-19 and planning interventions with a high impact on the disease itself and its ability to spread [12,17,18]. When a patient diagnosed with COVID-19 is admitted to an intensive care unit (ICU), they already present with impaired basic human needs. It is, therefore, important to adapt the provided care accordingly, to ensure its quality and to preserve the patient as a holistic being. To that purpose, it is paramount to organize the work processes within a nursing care logic [19].

According to Lima et al. (2021) “Nursing, as a professional area, when facing a patient infected with SARS-CoV-2, has the duty to ensure a high-quality practice, by exploring and empowering the available scientific tools, as well as by demonstrating a thorough preparation and deep knowledge of all the stages of the care process” [20]. With respect to the care process, the scientific knowledge and clinical reasoning that support nursing diagnoses are presented using a standardized language.

The nursing process, itself, is divided into five stages: assessment, diagnosis, planning, intervention, and evaluation [17,21]. This structure facilitates the nurses’ clinical reasonings, when caring for those in need (Swanson et al., 2021). However, in this exceptional situation, nurses encounter several challenges when seeking to fulfill patients’ needs specifically associated with the COVID-19 disease, since some of those needs are still unknown. It is, therefore, necessary to evaluate the nursing care’s effectiveness in this scenario, in order to optimize it (Swanson et al., 2021). It is in this context that this scoping review arises, as a way to solve the knowledge gap associated with human responses that are unknown, and in this way, build more adequate nursing processes.

The present review aims to identify the human responses exhibited by adult patients with COVID-19 by listing the corresponding formulated nursing diagnoses. In order to assess its relevance, a search was conducted in January 2021, via the following databases: JBI Evidence Synthesis, Cochrane, Medline, CINAHL Complete, and PubMed. Having verified that no similar work was published, the decision was made to perform a scoping review on the subject, following the recommendations provided in the latest manual published by the JBI in 2020 [22].

Research Question: based on the formulated nursing diagnoses, which human responses are exhibited by adult patients with COVID-19?

Inclusion Criteria: Population—All studies involving SARS-CoV-2-infected adult patients (aged 18 years or older) were considered. This, therefore, excluded any study with a population aged less than 18 years or that did not mention SARS-CoV-2 infections. Concept—All studies focusing directly or indirectly on nursing diagnoses, employing either the NANDA-I or the ICNP taxonomy, were considered. Context—All studies concerning the different contexts of nursing practices were considered, whether in hospital settings, community health, or home-based care.

Furthermore, all types of studies were considered whether of a qualitative or quantitative nature, including opinion articles and gray literature, as long as the works had been written in English, Portuguese, or Spanish, and made available between December 2019 and 15 January 2021.

## 2. Materials and Methods

This scoping review was conducted according to the recommendations provided by the Joanna Briggs Institute (JBI) [22].

### 2.1. Search Strategy

The search strategy was organized in three stages, as suggested in the latest manual published by the JBI. The first stage comprised a limited search on the PubMed and CINAHL Plus with Full Text databases. The keywords and index terms presented in each work’s title and abstract were analyzed, to obtain a search equation using Boolean operators and descriptors validated through the DeCS and MeSH platforms. After this formulation, the following search equation was designed: [(nursing diagnosis) OR (NANDA OR CIPE OR standardized nursing terminology OR standardised nursing language) AND (coronavirus OR covid-19 OR 2019-ncov OR COVID19 OR COVID-19 OR corona virus OR sars-cov-2)]. In the second stage, the search equation was applied to the following databases: CINAHL Complete (by EBSCO) Complementary Index (by B-on), MEDLINE (by B-on), Science Direct (by B-on), Academic Search Complete (by B-on), Science Citation Index (by B-on), Directory of Open Access Journals (by B-on), Scopus (by B-on), Supplemental Index (by B-on), Social Sciences Citation Index (by B-on), Business Source Complete (by B-on), Gale In Context Science (by B-on), Dialnet (by B-on), eBook Index (by B-on), ERIC (by B-on) and Cochrane Database of Systematic Reviews (by Cochrane Library). Finally, during the third stage, all selected bibliographic references were analyzed. The search was conducted between 14 and 31 January 2021. A table illustrating the search process employed in this review can be found in Appendix A Table A1.

### 2.2. Studies’ Selection

After finishing the search process, the obtained results were exported to Rayyan, an online systematic review analysis tool (Rayyan Systems, Inc., Doha, Qatar) and the selection process was performed using that platform. When completed, three independent reviewers analyzed the studies’ titles and abstracts and, later, performed full-text reading. Whenever an article’s selection raised doubts or uncertainty, its discussion was extended to a fourth reviewer. A PRISMA flow diagram depicting the result of the search process is presented in Figure 1.

### 2.3. Findings’ Presentation

The extracted findings were inserted into a data collection tool, using Microsoft Word™. This tool, created and validated by all reviewers, allowed for comparison and ordering of the various nursing diagnoses. The final sample was, thus, organized and characterized by the following categories: author(s), country, type of study, sample, objective(s), and nursing diagnoses—organized in accordance with the taxonomy used (either NANDA-I or ICNP). 

## 3. Results

### 3.1. Search Results

A total of 580 studies was initially identified, using the search equation described earlier; of these, only eight works complied with all the inclusion criteria and were effectively included. After analyzing their references, three more articles were considered for assessment, of which only one met the established criteria, thus leading to a total of 9 included studies. These were organized by author(s), publication year, and geographic distribution. With respect to the geographic distribution, there are three countries that stand out in this research field: Brazil (four articles); Spain (three studies); and USA (two works). This shows how this matter has become a worldwide concern, since these works are distributed by three continents. Furthermore, both the European and American continents present the largest number of produced studies on this topic.

### 3.2. Inclusion of Sources of Evidence

According to the levels of evidence established by Melnyk (Micah et al., 2020), this work includes: one level 1 study—an integrative literature review; two level 4 studies—case reports; three level 6 studies—qualitative works, 2 descriptive and 1 exploratory; three level 7 studies—expert opinions. The samples in these works are quite diverse; in some studies, the sample could be validated, while in others the sample had little significance or was not presented. The description of the characteristics of each included work is detailed in Appendix A Table A2.

### 3.3. Review’s Findings

The findings were organized to provide an overview on the subject at study. Using the data collection tool, the diagnoses were distributed according to the taxonomy employed in the analyzed studies: NANDA-I (*n* = 7), or ICNP (*n* = 1). In this regard, Table 1 presents the findings categorized in accordance with the ICNP’s taxonomy [23], while Table 2 lists the nursing diagnoses organized by the NANDA-I taxonomy II domains [5]. In Figure 2, it is possible to see a summary of Table 1 and in Figure 3 a summary of Table 2. Table 3 compares and demonstrates the main results of each taxonomy.

## 4. Discussion

The obtained data allowed identifying which human responses—based on the corresponding nursing diagnoses—were exhibited by SARS-CoV-2-infected adult patients. Regarding studies that applied the NANDA-I taxonomy, the most highlighted diagnoses were “impaired gas exchange” [2,17,20,24,25,26] and “ineffective breathing pattern” [2,17,20,24,25], followed by “ineffective protection” [2,17,24,25] and “hyperthermia” [2,17,20,24]. These diagnoses corroborate the following responses identified in adult individuals infected with the SARS-CoV-2 virus: cough, difficulty breathing, and fever [2]. On the other hand, the diagnoses presented below were less mentioned, despite being human responses also observed in SARS-CoV-2-infected patients: “risk for electrolyte imbalance” (associated with “vomiting”) [2,24], “diarrhea” [2,24] “activity intolerance” [17,24,25]. These responses typically emerge later in the course of the infection, so they are not usually observed during its onset [1,27]. The knowledge of these late human plans gave nurses the opportunity to build care plans focused on prevention and with greater vigilance.

Continuing our analysis of the findings pertaining to works that applied the NANDA-I taxonomy, “activity/rest” and “safety/protection” are the domains that contain human responses that correspond to the most prevalent diagnoses. Following these are the diagnoses belonging to the “health promotion” and “elimination and exchange” domains. The “activity/rest” domain focuses mainly on human responses associated with changes in the cardiovascular and pulmonary systems. The “safety/protection” domain addresses the preservation against losses and risks, as well as the absence of danger—either physical or physiological—and, in this specific case, the risk for infection and altered body thermoregulation, i.e., hyperthermia. The “health promotion” domain reflects the homeostasis in relation to the person’s balanced functions and well-being, namely to the harmonious control of health-related activities. The “elimination and exchange” domain focuses on diagnoses involving the secretion and excretion of organic products, including the production and exchange of gases, which connects it with the respiratory function [5].

Regarding studies that applied the ICNP taxonomy, the following focus correspond to the most frequent diagnoses: “dyspnea” [2,19], “cough” [2,19], “fever” [2,19], “vomiting” [2,19], “diarrhea” [2,19] and “hypoxia” [2,19]. In contrast, the following focus were less prevalent: “fatigue” [19], “social isolation” [19], “nasal discharge” [2], or “mechanical ventilation” [19].

Considering both taxonomies, the main diagnoses demonstrate an inadequate physiological ventilation [1,2,17,19,20,24,25], as well as an altered thermoregulation—namely on fever/hyperthermia [2,17,19,20,24]. Once again, we observed human responses very similar to NANDA’s Taxonomy, which allowed us to promote health, focusing on the person’s well-being and reducing anxiety related to the unknown.

Nursing care practice requires correct implementation of the nursing process, to systematize and organize the human responses identified by nurses (Souza et al., 2020). Thus, nursing diagnoses tend to focus on the assessment of vulnerabilities related to undesired human responses to living conditions, weaknesses, and intervention needs, among other factors. This requires the nurse to apply clinical reasoning, while assessing a patient, in order to formulate the correct diagnosis for the specific response presented by the patient [4,5]. In this sense, each person is regarded as a complex and ever-changing being, who requires assessment decentered from standardization, focused on the life process and health attitudes, both as an individual and as part of the community [5]. It is through this assessment that nurses diagnose the identified human response, which is the target of their intervention [4,5].

The study by González-Aguña et al. (2020) addressed this topic by cross-mapping among the NIC, NOC, and NANDA-I taxonomies, with subsequent validation by an expert panel applying the Delphi method [17]. They found that diagnoses, such as, “impaired spontaneous ventilation”, “ineffective breathing pattern”, “ineffective airway clearance”, and “risk for shock” were predominant, among others [17]. These findings were organized into the following categories: “Diagnoses in critical situation”, “Diagnoses focused on health outcome”, and “Diagnoses focused on outcome in iatrogenic prevention”. Moreover, according to the authors, regarding patients with COVID-19, it was clear the influence of their health status’—and, consequently, of their human responses’—evolution on multiple aspects of the provided care, given the disease’s unique complexity and variability [17].

The reflective study by Souza et al. (2020) used the ICNP taxonomy to identify the same diagnoses, through the corresponding human responses. These authors highlighted “fever”, “cough”, and “dyspnea” as the main triad in patients with COVID-19 [19]. Conversely, the worsening of the patients’ health-illness status increased the variety of nursing diagnoses, as we observed in the study by González-Aguña et al., with age and the existence of co-morbidities playing important roles in that regard [17,19]. As such, the admission of patients with COVID-19 to the ICU required close supervision, with increased and constant attention, in addition to demanding for new clinical assessments. In this regard, diagnoses, such as “hypertension/hypotension”, “vomiting”, “nausea”, “diarrhea”, “fatigue”, “impaired consciousness”, and “musculoskeletal pain” were observed in those ICU inpatients [19], but their clinical instability accentuated the progression towards more severe human responses, such as “hypoxia”, “mechanical ventilation”, “seizure”, and “risk for hemorrhage” [19]. In this scenario, the nurses’ surveillance was crucial and their ability to intervene early on was put to the test, since this infectious disease is quite recent and capable of producing numerous changes in those who are infected [19]. This study will make it possible to demonstrate knowledge capable of supporting the decision of the best care plan.

The study by Cussó, Navarro, and Gálvez (2020) focused on humanized care provided to adult patients with COVID-19. It was a case study about an 81-year-old elderly patient who went to an emergency department due to respiratory distress [26]. The case was analyzed from a holistic perspective, based on the humanistic nursing theory by Paterson and Zderad [28] and Watson’s theory of human care [29]. The authors mentioned that the emotional and spiritual support, inherent in nursing care, were hindered due to the use of personal protective equipment [26]. This difficulty allowed creating new strategies in the relationship with the person. These were the nursing diagnoses formulated during the initial observation, according to the NANDA-I taxonomy: “impaired gas exchange”, “risk for impaired skin integrity”, and “impaired urinary elimination” [26]. After 4 h, the diagnosis “risk for spiritual distress” was also added, due to the use of communication barriers, such as masks and protective suits, and to the absence of family visits, because of the restrictions imposed during the pandemic [26].

There are other studies that referred to emotional and spiritual suffering, highlighting diagnoses such as “fear” [25], “anxiety” [24,25], “death anxiety” [25], “spiritual distress” [2], and “impaired resilience” [25]. It is possible to observe that the promotion of spiritual well-being, as well as the prevention of loneliness and social isolation, were considered critical nursing assessments in avoiding psychological distress events, namely: anxiety, depression and post-traumatic stress. This corroborates the fact that mental health nursing has become an important intervention area, namely regarding the establishment of healthcare partnerships with the individual, the family, and/or the community, through humanized care [30].

Moorhead et al. (2021) and Swanson et al. (2021) analyzed the first human responses specifically directed towards the community. In a global perspective, it has become increasingly important to understand what the citizens’ human responses are, as a target group for care provision [18,25]. In this sense, there is a growing need to recognize the intrinsic vulnerabilities, problems, and risks of the world’s population, in order to encourage health-promoting behaviors. The pandemic forced the closure of public, social, cultural, and organizational institutions, with the consequent—and substantial—impact on family organization [25]. This, in turn, induced the surfacing of attitudes that endangered the community’s health, such as participating in public protests, gatherings, and social meetings during periods when these events would contribute to further spreading the SARS-CoV-2 virus [25]. The following correspond to diagnoses identified in these studies, with a set of community-related human responses: “deficient community health” [1,18], “ineffective community coping” [18], “risk for contamination” [18], “ineffective protection” [2,17,24,25], “risk-prone health behavior” [25] and “impaired comfort” [25].

Ramos et al. (2020) focused on the infectious disease impact on the community and highlighted two other important diagnoses: “risk for infection” and “social isolation” [1]. They related the first to the general population’s reduced knowledge about the exposure to microorganisms and associated the second with the need to remain quarantined and to keep social distancing, as part of the SARS-CoV-2 transmission control policy [1]. These clinical assessments draw attention to the importance of health promotion at a global level, in which nurses play a key role [1,25].

In summary, based on the corresponding nursing diagnoses and considering the NANDA-I, the ICNP, or taxonomy, the main human responses observed in adult patients with COVID-19 were related to fever/hyperthermia and changes in gas exchange or processes where the respiratory pattern was compromised. These coincided with the main symptoms affecting adult patients with COVID-19 [27].

Concerning the selected studies’ geographical distribution, this characteristic is likely influenced by the different levels of autonomy that nurses are allowed to have, worldwide. With respect to this matter, in several countries the characterization of human responses to COVID-19 has not been carried out using a standard language. This prevented the use of such information in data comparisons applying standard taxonomies and, consequently, its inclusion in this review.

## 5. Conclusions

Based on this review’s findings, the responsibility of nurses in the monitoring of adult patients with COVID-19, both in the community and in hospital settings, is clear since some of the identified human responses required nursing interventions.

Answering the research question, with respect to studies using the NANDA-I taxonomy, the findings have shown that, regarding those patients, “impaired gas exchange” was the most highlighted diagnosis [2,17,20,24,25,26], followed by “ineffective breathing pattern” [2,17,20,24,25]. Concurrently, regarding works employing the ICNP taxonomy, the relevant nursing diagnosis related to respiratory compromise was “cough present”, usually combined with “fever present”, “diarrhea present”, and “vomiting present” [2,19]. In view of these findings, we found that this disease causes physical responses associated with the respiratory system, so it will be easier to prevent them through the construction of adequate nursing interventions.

In addition, some of the analyzed works highlighted “risk for loneliness” and “social isolation” as important diagnoses [1,25], addressing them, alongside the “risk for infection”, from the community’s point of view. Concomitantly, other studies focused on human responses related to emotional and spiritual suffering, highlighting diagnoses, such as “fear”, “anxiety”, “death anxiety”, “spiritual suffering”, and “impaired resilience” [2,24,25,26]. Diagnosis assessment was considered important in the promotion of spiritual well-being, as well as in the prevention of loneliness and social isolation. They also demonstrated how transversal this topic was to several fields of nursing care, namely mental health nursing, here presented with community nursing [1,2,24,25]. In the future, it will be important to include the family in this disease process and appropriate interventions in patients’ mental health, as an intervention team. In addition to these areas, some of the selected studies also addressed human responses observed at the end-of-life, as well as in critically or chronically ill patients with COVID-19, whose illness aggravated.

While providing care to adult patients with COVID-19, nurses must be aware of each patient’s characteristics, and apply their scientific expertise and clinical reasoning to the greatest extent possible when formulating nursing diagnoses. Concurrently, as suggested by the human responses documented in this review, throughout the pandemic, the requirements for adequate care provision have been constantly updated to improve the quality of life of those patients, as much as possible.

In accordance with the NANDA-I taxonomy, a nursing diagnosis is “a clinical judgment concerning a human response to health conditions/life processes, or a vulnerability for that response, by an individual, family, group or community” [9]. In this regard, we consider that the present findings may contribute to the improvement of nursing care, by identifying foci of attention for nursing, within the current scenario.

One of the limitations of this review is that it only includes one study with level 1 evidence, according to Melnyk [22]. In this regard, we consider the further development of experimental studies vital to increase the available knowledge about this matter and to validate the human responses documented in this review. It will also be relevant to have studies that, based on this scoping review, find the best associated interventions to transfer this knowledge to practice and improve the quality of life of these patients.

## Figures and Tables

**Figure 1 ijerph-19-06332-f001:**
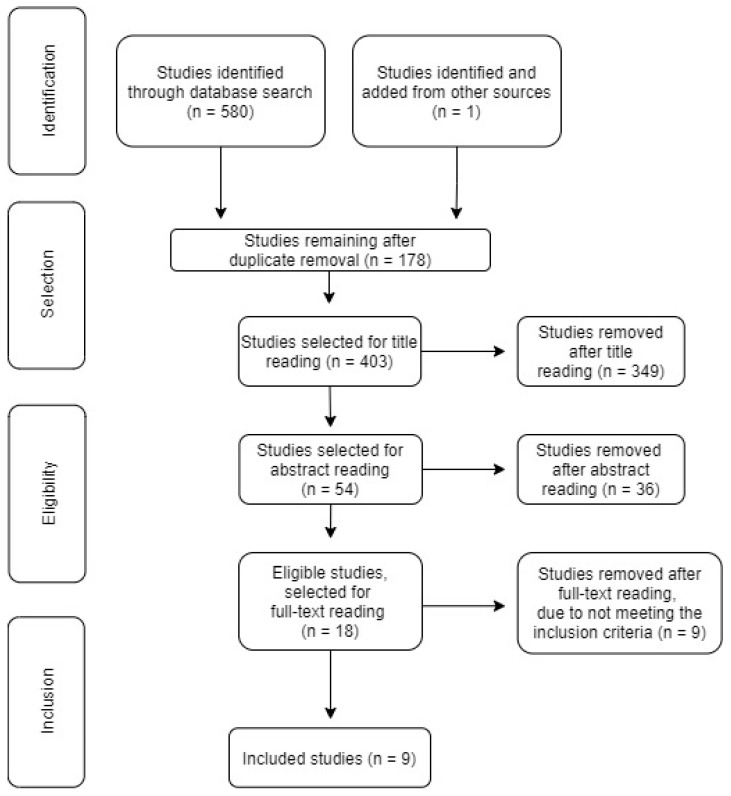
PRISMA flow diagram, adapted from the one proposed in the Joanna Briggs Institute’s methodological manual for scoping reviews [22].

**Figure 2 ijerph-19-06332-f002:**
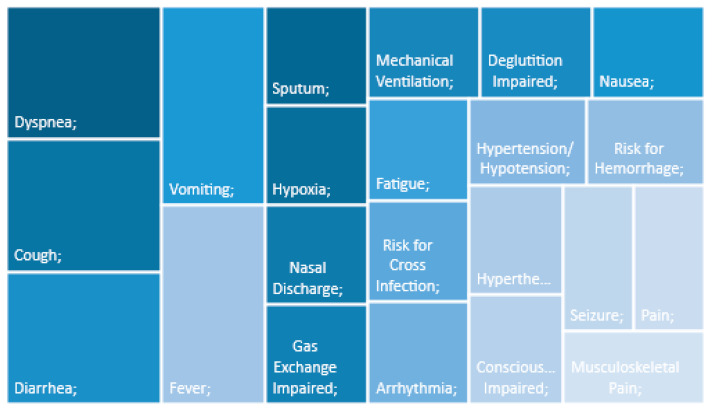
Nursing diagnoses categorization, by ICPN Focus [23].

**Figure 3 ijerph-19-06332-f003:**
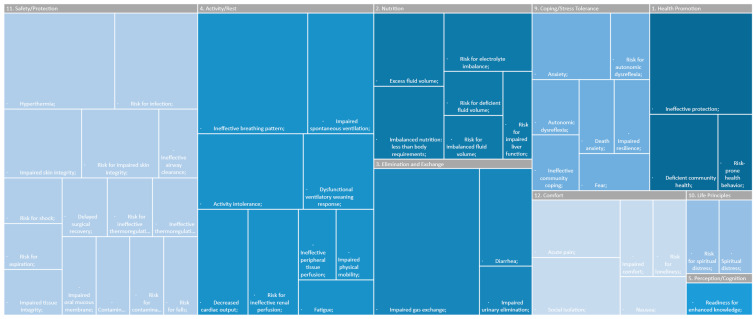
Nursing diagnoses categorization, by NANDA-I Taxonomy II Domains [5].

**Table 1 ijerph-19-06332-t001:** Nursing diagnoses categorization, by ICNP focus [23].

Focus	Judgement
Dyspnea [2,19]	(Undefined)
Sputum [19]	(Undefined)
Hypoxia [19]	(Undefined)
Cough [2,19]	(Undefined)
Nasal Discharge [2]	(Undefined)
Gas Exchange	Impaired [19]
Mechanical Ventilation [19]	(Undefined)
Deglutition	Impaired [19]
Diarrhea [2,19]	(Undefined)
Nausea [19]	(Undefined)
Vomiting [2,19]	(Undefined)
Fatigue [19]	(Undefined)
Cross Infection	Risk for [19]
Arrhythmia [19]	(Undefined)
Hypertension/Hypotension [19]	(Undefined)
Hemorrhage	Risk for [19]
Fever [2,19]	(Undefined)
Hyperthermia [2]	(Undefined)
Consciousness	Impaired [19]
Seizure [19]	(Undefined)
Pain [2]	(Undefined)
Musculoskeletal Pain [19]	(Undefined)
Social Isolation [19]	(Undefined)
Edema (Lower Limbs) [2]	(Undefined)

**Table 2 ijerph-19-06332-t002:** Nursing diagnoses categorization, by NANDA-I taxonomy II domains [5].

NANDA-I Taxonomy II Domains	Nursing Diagnoses
1. Health Promotion	Ineffective protection [2,17,24,25]; Deficient community health [1,18]; Risk-prone health behavior [25]
2. Nutrition	Risk for deficient fluid volume [17] Risk for imbalanced fluid volume [17] Excess fluid volume [2,17]; Imbalanced nutrition: less than body requirements [17,26]; Risk for impaired liver function [17] Risk for electrolyte imbalance [2,26].
3. Elimination and Exchange	Impaired gas exchange [2,17,20,24,25,26] Impaired urinary elimination [26]; Diarrhea [2,26].
4. Activity/Rest	Dysfunctional ventilatory weaning response [17,25]; Decreased cardiac output [2,17]; Ineffective breathing pattern [2,17,20,24,25]; Ineffective peripheral tissue perfusion [17]; Impaired spontaneous ventilation [2,17,25]; Activity intolerance [17,25,26]; Impaired physical mobility [17]; Risk for ineffective renal perfusion [withdrawn] [17,26]; Fatigue [2].
5. Perception/Cognition	Readiness for enhanced knowledge [26];
6. Self-Perception	Unmentioned
7. Role Relationships	Unmentioned
8. Sexuality	Unmentioned
9. Coping/Stress Tolerance	Risk for autonomic dysreflexia [17]; Autonomic dysreflexia [17]; Ineffective community coping [18]; Anxiety [24,25]; Death anxiety [25]; Impaired resilience [25]; Fear [25].
10. Life Principles	Risk for spiritual distress [26]; Spiritual distress [2].
11. Safety/Protection	Ineffective airway clearance [17]; Risk for infection [1,17,25]; Risk for shock [17]; Impaired skin integrity [17,26]; Risk for aspiration [17]; Risk for impaired skin integrity [17,26]; Hyperthermia [2,17,20,26]; Impaired tissue integrity [17]; Delayed surgical recovery [17]; Risk for ineffective thermoregulation [17]; Ineffective thermoregulation [17]; Impaired oral mucous membrane [17]; Contamination [18]; Risk for contamination [18]; Risk for falls [26].
12. Comfort	Acute pain [2,17]; Impaired comfort [25]; Social isolation [1,25]; Risk for loneliness [25]; Nausea [2].
13. Growth/Development	Unmentioned

**Table 3 ijerph-19-06332-t003:** Compares and demonstrates the main results of each taxonomy.

Nursing Diagnosis	
NANDA-I Taxonomy II Domains	ICNP’s Taxonomy
Impaired gas exchange [2,17,20,24,25,26] Ineffective breathing pattern [2,17,20,24,25] Ineffective protection [2,17,24,25]; Hyperthermia [2,17,20,26]	Dyspnea [2,19] Cough [2,19] Fever [2,19] Vomiting [2,19] Diarrhea [2,19] Hypoxia [2,19]

## Data Availability

Given the nature of the data that constitute the corpus of this review article, there were no archived datasets since the data that make up the review come from articles that have already been published. The original articles are properly referenced in the list of references.

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
