# Peer review of "Responses Presented by Adult Patients with COVID-19, Based on the Formulated Nursing Diagnoses: A Scoping Review"

_ijerph, 2022, doi:10.3390/ijerph19106332_

Round 1

Reviewer 1 Report

Dear authors,

It was a pleasure for me to read your article. Especially the method you use and the results you find are scientifically valuable. For this reason, I congratulate you. However, there are issues that need to be improved in various parts of the article. I think doing this will increase the quality of your article. My recommendations are presented below.

I wish you good work.

Kind regards.

In the article, I suggest developing the following aspects:

  1. In the abstract, it should be stated between which dates the research covers and which databases are included in the research.
  2. In the introduction, the gap that this research will fill in the literature should be presented more clearly and convincingly. Namely; the subject of the research was stated and the importance of nursing diagnosis was mentioned. However, as the special subject of this study, the contribution of the responses presented by adult patients with COVID-19 to the literature should be explained more clearly.
  3. The information given in the table in the discussion section is presented according to NANDA-I or ICNP taxonomy. The Discussion section, as it stands, gives the appearance of presenting summaries of these studies. For this reason, making comments after the research results are given in the discussion section and making comparisons with the results of previous research based on these comments will increase the quality of the article. Another issue that will increase the quality of the article is to present a graphic that will reveal the cross-links between the results of these researches made according to two different taxonomies.
  4. The conclusion section should not be designed as a short summary of the findings in the article, but as a section where recommendations based on these findings are given. Therefore, it is important to rewrite the conclusion section and associate it with the results by presenting more suggestions.

Author Response

Good night,

I would like to start by thanking you for your enriching comments on my article.

So now I'm going to respond to your comments:

  1. By mistake we did not introduce the date of the research and the databases used but we have already updated this in the attached document;
  2. We corrected the introduction and I think that at this point the idea you suggested is clearly in the text;
  3. As suggested, we made some comments in the article's discussion. As for the comparison with past studies, it becomes complicated because the topic under study emerged in the time frame included in the research.As for the graphs, we chose to present the result in this way and put the tables in an appendix, I would like you to give me feedback if this construction adds value to the article. We also created a table with the most relevant results.

  1. Based on your suggestion, we rectify the conclusion.

I await your response and I am available for any corrections that you think are necessary.

Kind regards,

Reviewer 2 Report

Dear Authors,
thank you for the opportunity to read your manuscript.

The more we know about this disease, as you rightly point out, the easier it is for nurses to make accurate diagnoses and select appropriate interventions. As you can see from your revision, whether we use NANDA or ICNP diagnoses, we get the same result. Your article has another additional value - no matter which typology of diagnoses is used, the same result is always obtained. Demonstrating the superiority of one typology over another clearly does not make much sense. In the end, it is always about the well-being of the person. 
You have planned and carried out the analysis very well. However, I miss the practical translation, after all nursing is a practical discipline. 

Author Response

Good night,

First of all, thank you for your appreciation of my article.

Upon reading your comment, I understood your question perfectly, not least because I am also a nurse and there is an increasing need for research that promotes a more evidence-based practice, in order to increase the quality of care.

As a way to respond, I tried to issue some comments and suggestions throughout the article, inviting you to read our article again.

I ask that if you continue to feel this restlessness, contact us so that we can rectify it in the best way possible.

Kind regards,
